# Quality of Life and Loneliness Among Older Adults in Primorsko-Goranska County

**DOI:** 10.3390/ijerph22111713

**Published:** 2025-11-13

**Authors:** Laura Jagić, Katarina Galof, Željko Jovanović, Bojan Miletić, Marija Spevan

**Affiliations:** 1Faculty of Health Studies, University of Rijeka, 51000 Rijeka, Croatia; ljagic@student.uniri.hr (L.J.); bojan.miletic@uniri.hr (B.M.); marija.spevan@uniri.hr (M.S.); 2Faculty of Health Sciences, University of Ljubljana, 1000 Ljubljana, Slovenia; katarina.galof@zf.uni-lj.si

**Keywords:** quality of life, ageing, elderly, old age, loneliness

## Abstract

*Background:* Ageing is accompanied by physical, psychological and social changes that can negatively affect the quality of life of older adults. Loneliness is one of the most important psychosocial problems in later life and is closely related to lower life satisfaction. The aim of this study was to investigate the relationship between subjective loneliness and quality of life in older adults in Primorje-Gorski Kotar County, taking into account gender and living arrangements. *Methods:* A convenience sample of 153 adults aged 63 years and older participated in the study. Quality of life was assessed using the Personal Well-Being Index-Adult (PWI-A), and loneliness was measured using the short version of the UCLA Loneliness Scale. Statistical analyses included descriptive statistics, Spearman correlation and Mann–Whitney U test, performed in Statistica 14.0. *Results:* A significant negative correlation was found between loneliness and quality of life (r = −0.448; *p* < 0.01). A significant negative correlation was found between loneliness and quality of life (r = −0.448; *p* < 0.01). Women reported significantly higher levels of subjective well-being than men (*p* < 0.05), while no significant gender difference was observed for loneliness. No significant differences were found based on living arrangement (living alone vs. with others) (*p* > 0.05). *Conclusions:* The results confirm that loneliness significantly affects the subjective well-being of older adults. Gender differences were observed in subjective well-being but not in loneliness. Living arrangement did not show a strong influence. These results emphasize the need for targeted strategies and psychosocial interventions aimed at reducing loneliness and improving quality of life in ageing populations.

## 1. Introduction

Old age represents the final stage of human development and can be defined through several complementary criteria. Chronologically, it typically refers to adults aged 65 and above; socially, it is associated with changes in roles and status, especially following retirement; and functionally, it is characterized by noticeable declines in physical and cognitive abilities [1]. These transitions are often accompanied by retirement, loss of social roles, reduced mobility, and the death of significant others, which may collectively reduce social participation and negatively affect perceived quality of life (QoL) [2].

Ageing is a dynamic and multidimensional process characterized by biological, psychological, and social changes. Biological ageing refers to the gradual decline in physiological functions; psychological ageing includes changes in cognitive and emotional functioning; and social ageing refers to alterations in a person’s interaction with their environment and social roles. In addition to these functional and structural changes, ageing also entails psychosocial adjustments. The concept of ageing is therefore not limited to biological decline but includes the evolving relationship between the individual and their social environment. This multidimensional nature of ageing underscores the necessity of considering both objective and subjective indicators—such as physical health, autonomy, and perceived well-being—when assessing QoL in later life [3].

The loss of meaningful life roles, decreased social ties, and limited social activity can lead to social isolation and subsequent feelings of loneliness among older adults. Loneliness has profound effects on both physical and psychological well-being. It is associated with memory loss, unhappiness, low self-esteem, depression, hypertension, and other chronic illnesses, all of which contribute to a reduced quality of life and increased mortality. Negative emotions associated with loneliness often arise from a discrepancy between the desired and perceived quality and quantity of social relationships or from a subjective sense of social isolation and a lack of meaningful social connection [3].

A recent meta-analysis by Sulandari et al. confirmed that loneliness is strongly associated with lower life satisfaction and poorer QoL in older adults, underscoring its importance as a public health concern [4]. Similarly, a global meta-analysis involving more than 1.25 million older adults found a loneliness prevalence rate of 27.6%, with loneliness significantly associated with reduced QoL and lower levels of social support [5]. Moreover, a longitudinal meta-analysis revealed that loneliness increases the risk of all-cause mortality (HR = 1.14; 95% CI 1.10–1.18), highlighting its detrimental effects on both survival and well-being [6]. A systematic literature review on community-dwelling older adults further emphasizes that loneliness is linked to reductions in subjective well-being, with different forms of loneliness yielding distinct adverse outcomes [4,5,6,7].

The subjective experience of loneliness is strongly linked to perceived QoL. Quality of life is a complex, multidimensional concept encompassing both subjective evaluations and objective living conditions. It generally refers to a person’s assessment of their circumstances in interaction with their physical and social environment. This evaluation is shaped by individual characteristics such as personality traits, attitudes, values, and beliefs, and reflects one’s unique perception of well-being and satisfaction [8]. Objective indicators of QoL include economic stability, housing conditions, access to healthcare, and social support systems. Hence, QoL can be broadly defined as the subjective appraisal of how external conditions affect personal life [9].

### 1.1. Quality of Life in Old Age: A Multidimensional Concept and Its Challenges

Quality of life among older adults is defined as a multidimensional construct encompassing physical health, psychological well-being, independence, social relationships, personal beliefs, and the living environment [8]. From an occupational and social health perspective, QoL reflects an individual’s capacity to maintain autonomy, meaningful engagement, and emotional stability. Previous research has shown that decreased social contact and reduced participation in meaningful activities are among the most prominent predictors of lower QoL in later life [10,11].

In institutional care settings, QoL is largely dependent on interpersonal relationships with staff, participation in decision-making, and access to meaningful activities that maintain a person’s dignity and identity [10]. For older adults living in the community, life satisfaction is enhanced by strong interpersonal relationships, social participation and maintaining independence. In addition, factors such as gender, education level and marital status have been shown to influence perceptions of life satisfaction in old age [11].

### 1.2. Loneliness as a Challenge for Public Health

Loneliness is a growing problem in the older population and is considered one of the most important predictors of reduced psychosocial well-being. A distinction is made between emotional loneliness, which refers to the lack of close emotional ties, and social loneliness, which refers to the lack of broader social networks. Loneliness is usually described as a distressing experience resulting from the perceived inadequacy of social relationships and serves as a subjective indicator of an unmet need for social support [12].

According to research by the United Nations Population Fund (UNFPA), factors such as being alone, deteriorating health, loss of a partner and social exclusion contribute significantly to loneliness in old age [13]. One of the key psychosocial factors that undermines QoL in older age is loneliness. Loneliness can be understood as a subjective feeling of social isolation resulting from a discrepancy between desired and actual social relationships [12]. It is associated with a range of adverse outcomes, including depression, cognitive decline, cardiovascular disease, and higher mortality [13,14]. As such, loneliness is increasingly recognized as a pressing global public health challenge, comparable in its impact to smoking or physical inactivity [15].

### 1.3. The Relationship Between Loneliness and Quality of Life

A growing body of research confirms a strong, bidirectional relationship between loneliness and QoL in old age. Older adults who report higher levels of loneliness tend to have lower levels of life satisfaction, emotional stability and optimism, as well as a more pessimistic outlook for the future [13]. Conversely, a reduced QoL—particularly if it is due to poor health, financial insecurity and social exclusion—can exacerbate feelings of loneliness.

Correlational and longitudinal studies have shown that persistent loneliness is an important predictor of declining QoL in old age, particularly in the absence of adequate social support networks. The UNFPA [13] has reported that older adults with higher levels of social isolation consistently score lower on all dimensions of QoL than their socially engaged peers. In response, a wide range of interventions focusing on social integration and emotional support have been developed to address this issue. These initiatives not only enhance individual well-being but also alleviate pressure on health and social care systems [16]. However, the academic value of the present study lies in its regional and contextual focus. Older adults in Primorsko-Goranska County represent a distinct population with specific socio-demographic, cultural, and economic characteristics that differ from those examined in large-scale international studies. The study contributes new insight into how cultural patterns, social support systems, and regional living conditions influence the relationship between loneliness and quality of life. This local perspective offers valuable comparative evidence for the broader academic community studying ageing, loneliness, and well-being.

Therefore, the present study offers an innovative contribution by investigating the relationship between subjective loneliness and QoL in older adults in Primorje-Gorski Kotar County, considering gender and living arrangements. By situating this analysis in a specific sociocultural environment, the study aims to enhance understanding of how local factors shape psychosocial well-being in older age and to provide evidence-based insights relevant to occupational and community health interventions.

### 1.4. Purpose and Objective

The main objective of this study was to investigate the relationship between the subjective experience of loneliness and the quality of life of older adults in Primorje-Gorski Kotar County. Primorje-Gorski Kotar County is a multicultural region of Croatia with both urban and rural areas, and a relatively high proportion of older adults, partly due to retirement migration. This demographic context makes it a relevant setting for studying well-being and loneliness in ageing populations.

The specific objectives were as follows:-To investigate the influence of loneliness on the assessment of QoL in older adults;-To investigate the experience of loneliness and QoL in older adults as a function of gender;-To examine the experience of loneliness and QoL in older adults as a function of the living arrangements in which they live.

Based on the aims, the following hypotheses were formulated:

**H1.** 
*There is a significant negative correlation between loneliness and quality of life in older adults.*


**H2.** *Women will report higher levels of loneliness and lower quality of life compared to men*.

**H3.** *Older adults living alone will report higher levels of loneliness and lower quality of life compared to those living with others*.

The novelty of this study lies in its focus on a demographically unique region, combining global theoretical insights with locally grounded data to illuminate how cultural and social contexts shape the experience of loneliness and quality of life among older adults.

## 2. Materials and Methods

### 2.1. Participants

The study involved 153 males and females aged 63 years and older in Primorje-Gorski Kotar County. The participants were recruited using a convenience sampling method through local pensioners’ clubs and health centres. Sample size: No a priori power analysis was performed, but the final sample (N = 153) is comparable to similar studies in the field. Missing data: Minor discrepancies in the number of participants across analyses reflect incomplete responses on some items, which precluded calculation of composite scores in those cases. The inclusion criteria were persons living in their own household, able to care for themselves and able to perform activities of daily living. The exclusion criteria included mental disorders and physical illnesses that affect the ageing process, such as Alzheimer’s disease.

### 2.2. Procedure

Data were collected using the paper-and-pencil method. The questionnaire included the Personal Well-Being Index-Adult (PWI-A) and the short form of the UCLA Loneliness Scale [17,18]. The PWI-A and the short form of the UCLA Loneliness Scale were administered in validated Croatian translations, published and used in prior research. The distribution of the questionnaire was supported by the Belveder-Kozala Pensioners’ Club, the Čavle Pensioners’ Club, the Zamet Pensioners’ Club, the Orehovica Health Centre and the Kozala Health Centre.

### 2.3. Measurements

#### 2.3.1. Personal Well-Being Index for Adults (PWI-A)

The Personal Well-Being Index (PWI-A) can consist of either seven or eight questions. The standard version contains seven core questions relating to satisfaction with different areas of life. The PWI-A scale, which is available for academic use, consists of seven items that assess satisfaction with QoL in the following areas: health, standard of living, life achievements, relationships with family and friends, personal security, sense of belonging to the community and future security. Respondents are asked: “How satisfied are you with …?”.

The answers are given on an 11-point scale (from 0 to 10), where 0 stands for “completely dissatisfied” and 10 for “completely satisfied”. The index may only be completed by the respondent him/herself. It is not permitted for another person to fill in the questions for the respondent. The average value of all areas results in the PWI-A [17,19].

#### 2.3.2. UCLA Loneliness Scale (UCLA Scale)

Both unidimensional and multidimensional scales are used to measure loneliness. More specifically, several instruments have been developed based on two main criteria: (1) the duration of loneliness and (2) the dimensionality of loneliness. In terms of duration, the New York University Loneliness Scale (NYU), developed by Rubinstein and Shaver, measures loneliness as a personality trait. The most used instrument is the UCLA Loneliness Scale [20], which measures global loneliness as a state. However, numerous studies have shown that the scale is not unifactorial, as the number of factors varies in different samples. In addition, the results on gender differences in loneliness are inconsistent across different groups. Therefore, Allen and Oshagan [21] proposed a short form of the UCLA scale consisting of seven items.

The short form of the UCLA scale, which is also available for academic use, was developed to measure the subjective experience of loneliness and social isolation. The scale consists of seven items, which are answered on a five-point Likert scale, where 1 means “does not apply to me at all” and 5 means “applies to me completely”. The score is calculated from the sum of the scores of all items divided by the number of items. A total score above three indicates a higher level of perceived loneliness, while a score below three reflects a lower level of perceived loneliness [18,22]. In this study, consistent with validation recommendations, we report the mean item score (range 1–5) rather than a total summed score.

### 2.4. Statistical Analyses

Descriptive statistics (frequencies, percentages, means and standard deviations) were used to illustrate the basic characteristics. Normality was first assessed using the Kolmogorov–Smirnov and Shapiro–Wilk tests. As the data did not meet the assumptions of normality, non-parametric tests were used. Spearman’s rank correlation coefficient was employed to analyze the relationship between loneliness and QoL. Mann–Whitney U tests were used to compare QoL and loneliness by gender and living arrangement. Statistical significance was set at *p* < 0.05. Data were analyzed using Statistica version 14.0.0.15 [23]. Reliability: In our sample, internal consistency was satisfactory (PWI-A α = 0.85; UCLA short form α = 0.81). This approach allows a precise and objective assessment of the impact of loneliness on the QoL of older adults and the investigation of group differences.

## 3. Results

### 3.1. Socio-Demographic Characteristics of the Participants

The data on the gender distribution of the 153 participants show that 69.9% were female and 30.1% male (Table 1). In response to the question “Who do you live with?”, 33.3% of participants stated that they lived alone, while 66.7% stated that they lived with household members (such as partners, children or other family members). The mean age of participants was 75.32 years with a standard deviation of 7.226. The youngest participant was 63 years old, while the oldest was 96 years old.

These characteristics provide a context for understanding the sample and may be important in interpreting differences in subjective well-being and loneliness in subsequent analyses.

The highest mean scores in Table 2 for the PWI-A responses were recorded for the item “How satisfied are you with your personal relationships?” where the mean score was 8.22 with a standard deviation of 1.88, followed by “How satisfied are you with what you achieve in life?” with a mean score of 7.97 and a standard deviation of 1.80. The lowest mean response values were observed for the item “How satisfied are you with your health?” with a mean value of 6.03 and a standard deviation of 2.34, followed by “How satisfied are you with your future security?” with a mean value of 6.99 and a standard deviation of 2.09.

The highest mean score for UCLA in Table 2 was for the item “I do not share my thoughts and ideas with others”, with a mean score of 2.77 and a standard deviation of 1.41. The lowest mean response value was obtained for the item “I am unhappy because I withdraw so much” with a mean of 1.90 and a standard deviation of 1.21. The mean value of the overall UCLA score was 2.52 ± 1.33, indicating a moderate level of loneliness among participants.

The data obtained for the observed factors show that the mean score of the PWI-A scale is 74.46, with a standard deviation of 15.40, while the mean item score on the UCLA scale was 2.52 (SD = 1.33).

According to the categories of the PWI-A questionnaire, 6.7% (N = 10) of the 150 respondents reported a “challenged” level of subjective well-being, 28.7% (N = 43) reported an “impaired” level, while 64.7% (N = 97) reported a “normal” level of subjective well-being.

When analyzing the significance level for the questions “How satisfied are you with how safe you feel?”, “How satisfied are you with your sense of belonging to the community?” and “I do not share my thoughts and ideas with others”, the significance value of the chi-square test is *p*1 < 0.05.

For the questions “Nobody knows me well” and “Adults are around me, but not with me”, the chi-square test also shows a significance value of *p*2 < 0.05, which indicates a statistically significant difference in relation to the participants’ circumstances.

According to the significance values in Table 3, a *p*-value of more than 0.05 (*p* > 0.05) indicates that the data follows a normal distribution, while a *p*-value of less than 0.05 (*p* < 0.05) indicates a deviation from normality. As the significance level in the Shapiro–Wilk test is not above 0.05 for either scale, the normal distribution was not confirmed. Therefore, non-parametric statistical methods were used in the subsequent analysis.

### 3.2. Correlation Analysis

To analyze the relationship between subjective well-being and perceived loneliness, Spearman’s rank correlation coefficient was calculated. A moderate negative correlation was found between PWI-A and UCLA scores (r = −0.448, *p* < 0.01), suggesting that higher levels of subjective well-being are associated with lower levels of perceived loneliness (Figure 1).

### 3.3. Differences in Subjective Well-Being and Loneliness by Gender and Living Arrangements

The Mann–Whitney U-test was used to analyze the differences in subjective well-being (PWI-A) and loneliness (UCLA) according to gender and living arrangements. This non-parametric test was used due to the lack of normal distribution of the data (Table 3).

Using the significance value for the PWI-A variable (Table 4), it was found that *p* < 0.05, indicating that women scored significantly higher on subjective well-being than men (*p* = 0.036). For loneliness, the gender difference was not statistically significant. The rank analysis showed that females had a higher mean rank, suggesting that their ratings for the observed factor were higher compared to the male participants.

### 3.4. Differences in Subjective Well-Being and Loneliness According to Living Arrangements

When examining the significant values for the analysis’s factors depending on who the participants live with (Table 5), it was found that there was no statistically significant difference in the responses based on living arrangements (*p* > 0.05).

## 4. Discussion

In this study, subjective well-being was conceptualized as a core component of overall quality of life. Our findings confirm a significant negative association between loneliness and subjective well-being among older adults, consistent with prior research identifying loneliness as one of the strongest predictors of reduced QoL and life satisfaction in later life [24,25,26,27]. Specifically, the negative correlation between the PWI-A and the UCLA Loneliness Scale (r = −0.448; *p* < 0.01) aligns with international studies that emphasize loneliness as a central determinant of older adults’ psychological well-being.

Although this study operates QoL using the Personal Well-Being Index (PWI-A), the results are in accordance with the broader concept of QoL, understood as a multidimensional construct that encompasses subjective well-being, satisfaction, and perceived control. In this context, subjective well-being is interpreted as a core dimension of QoL rather than a separate construct.

Unlike most previous studies, the present research was conducted in Primorje-Gorski Kotar County—a demographically unique region characterized by a multicultural population, both urban and rural areas, and a high proportion of older adults due to retirement migration. This regional and cultural context provides valuable insights into how local factors, such as family dynamics, community engagement, and intergenerational support, influence the experience of loneliness and overall QoL.

Interestingly, women in our sample reported higher subjective well-being than men, while no significant gender differences in loneliness were observed. This partially contrasts with findings from other cultural contexts, such as Sri Lanka [28] and China [29], which reported greater loneliness among women. The divergence may reflect gender-specific coping strategies or the influence of Croatian social norms, where women often maintain stronger emotional and community ties that mitigate the negative impact of loneliness on well-being.

Regarding living arrangements, no significant differences were found in either loneliness or QoL. This suggests that the quality of social relationships—rather than household composition alone—is the more decisive factor for well-being. Previous research by Tourani et al. [27], by Kousha et al. [30], and Shpakou et al. [31] similarly highlighted that emotional connectedness and satisfaction with relationships exert a stronger influence on psychological well-being than mere physical cohabitation. These results emphasize that interventions should focus on strengthening meaningful social relationships and community participation rather than exclusively addressing living arrangements.

Cultural context plays an important role in explaining such findings. In Mediterranean and Central European societies, including Croatia, older adults often maintain close contact with family members and neighbours, which may buffer the effects of living alone. In contrast, older adults in more individualistic societies may rely more on institutional or digital forms of support, which could intensify loneliness. This supports the growing understanding that loneliness is shaped not only by personal characteristics but also by broader cultural and social structures.

Moreover, prior studies by Xie et al. [29] demonstrated that subjective age—how old one feels—affects loneliness through mediating variables such as resilience and self-esteem. These findings are conceptually consistent with our results, which indicate that satisfaction with social relationships contributes positively to psychological well-being. Similarly, Attafuah et al. [32] and Stevens [33] found that accessible healthcare, financial independence, religious engagement, and strong family support all enhance QoL, particularly in low-resource settings.

Loneliness should therefore be recognized as a multidimensional and cumulative phenomenon—not merely the absence of social contact, but also dissatisfaction with existing relationships and the perceived loss of meaningful roles in society. It constitutes not only a psychosocial concern but also a major health risk, as it has been linked to cardiovascular disease, cognitive decline, sleep disorders, and increased mortality [34,35,36]. Recognizing loneliness as a public health issue rather than a private emotional experience is crucial for developing effective prevention and intervention strategies.

Limitations and practical implications: This study has several limitations. The data were collected using self-report instruments and a convenience sample from a single region, which may limit external generalizability. Regression or longitudinal analyses were not conducted; such approaches could further clarify interactions between gender, age, and living arrangements. Moreover, the exclusion of participants with cognitive impairments was based on self-report rather than standardized cognitive testing, which may have introduced bias. Future studies should incorporate validated tools such as the Mini-Mental State Examination (MMSE) to ensure more precise screening. Due to the non-normal distribution of several variables, non-parametric statistical methods were used, which may reduce the sensitivity of the analyses [37]. Because the study was limited to one Croatian county, the findings may not be generalizable to other cultural or national contexts. Despite these limitations, the study offers meaningful insights into the relationship between loneliness and QoL among older adults in a specific sociocultural context. The results highlight the importance of social connectedness, emotional support, and resilience as key protective factors.

From a practical perspective, these findings emphasize the need for community-based interventions that promote social participation, intergenerational programmes, and digital inclusion among older adults. Occupational therapists, social workers, and community health professionals should collaborate to design and implement programmes that enhance engagement, self-efficacy, and meaningful daily activities, thereby improving the quality of life in later years.

## 5. Conclusions

Loneliness has a significant negative impact on the subjective assessment of QoL in older adults. Gender differences were found only for subjective well-being, with women reporting higher well-being than men, while no significant gender differences were found for loneliness. Living arrangement did not significantly influence either outcome, contrary to our original hypothesis that people living alone would report higher levels of loneliness and lower perceived QoL. These findings suggest that the quality of social relationships, rather than the mere presence of cohabitation, may be the key determinant of well-being in later life. Despite these limitations, the study makes an important contribution to understanding the complex relationship between loneliness and QoL in older adults. By combining global theoretical perspectives with locally grounded data, it emphasizes the need for culturally sensitive, gender-responsive, and holistic approaches to addressing loneliness in ageing populations.

The UCLA scale results are now presented consistently as mean values per item, clarifying previous ambiguities; however, due to the lack of regression analyses, the interaction between gender, living arrangements, and loneliness remains to be explored. Finally, cultural factors specific to the local context may limit the transferability of the results to broader populations.

Nevertheless, this study makes an important contribution to understanding the impact of loneliness on older adults’ QoL. By integrating global perspectives with locally anchored data, it emphasizes the importance of culturally sensitive, gender-specific and holistic approaches to tackling loneliness in older populations.

Future research should employ larger and more representative samples to enhance external validity. Longitudinal designs would be particularly valuable for tracking changes over time and identifying causal relationships. Moreover, future studies should examine the role of protective psychosocial factors—such as resilience, self-esteem, and social support networks—as potential moderators of loneliness and well-being. Intervention-based studies are also needed to evaluate the effectiveness of targeted programmes aimed at reducing loneliness and improving subjective well-being, especially among older adults living alone or in institutional care. Particular attention should be given to fostering intergenerational relationships, promoting resilience, and reducing the stigma associated with ageing and loneliness.

## Figures and Tables

**Figure 1 ijerph-22-01713-f001:**
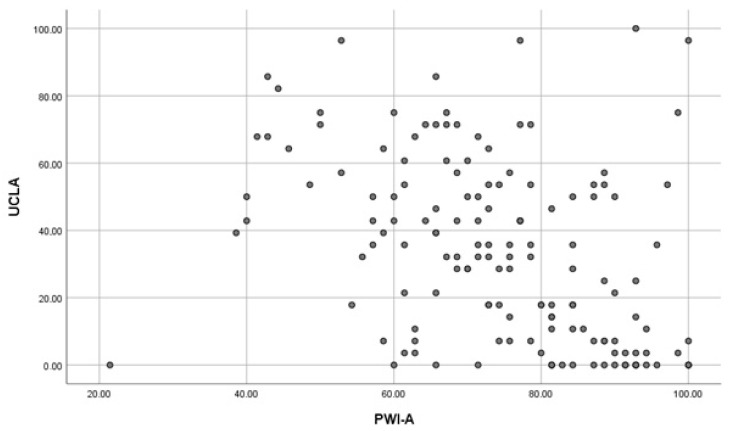
Scatterplot of the correlation between PWI-A and UCLA.

**Table 1 ijerph-22-01713-t001:** The socio-demographic characteristics of the study participants.

Variable	Category	N	%	Mean	SD	Min	Max
Gender	Female	107	69.0				
	Male	46	30.1				
Living arrangements	Living alone	51	33.3				
	With household members	102	66.7				
Age		153	100.0	75.32	7.23	63	96

**Table 2 ijerph-22-01713-t002:** Results of the PWI-A and UCLA scale.

Measurement	Item	N	Nf	Nm	$\mathrm{Mean}\;(\bar{x})$	SD	*p*1*	*p*2*
PWI-A	How satisfied are you with your standard of living?	153	107	46	7.23	1.94	0.469	0.252
PWI-A	How satisfied are you with your health?	153	107	46	6.03	2.34	0.646	0.641
PWI-A	How satisfied are you with what you have achieved in life?	153	107	46	7.97	1.80	0.140	0.197
PWI-A	How satisfied are you with your personal relationships?	152	106	46	8.22	1.88	0.598	0.744
PWI-A	How satisfied are you with how safe you feel?	152	106	46	7.62	1.88	0.038	0.139
PWI-A	How satisfied are you with the feeling of being part of your community?	153	107	46	7.79	1.97	0.003	0.884
PWI-A	How satisfied are you with your future security?	152	107	45	6.99	2.09	0.201	0.576
UCLA	I feel a lack of companionship.	152	106	46	2.39	1.42	0.390	0.061
UCLA	I don’t feel close to anyone.	153	107	46	2.28	1.38	0.857	0.123
UCLA	I do not share my thoughts and ideas with others.	153	107	46	2.77	1.41	0.005	0.097
UCLA	Nobody knows me really well.	152	106	46	2.58	1.36	0.424	0.024
UCLA	My social relationships are superficial.	153	107	46	2.31	1.27	0.662	0.889
UCLA	I am unhappy because I am so withdrawn.	153	107	46	1.90	1.21	0.769	0.680
UCLA	There are adults around me, but not with me.	152	106	46	2.41	1.44	0.354	0.027

N—number of participants, Nf—number of women, Nm—number of men, mean value ($\bar{x}$), SD—standard deviation, *p*1*—significance level gender, *p*2*—significance level living arrangements.

**Table 3 ijerph-22-01713-t003:** Normality test of the distribution.

Scala	Kolmogorov–Smirnov Statistic	df	Sig.	Shapiro–Wilk Statistic	df	Sig.
PWI-A	0.056	148	0.200 *	0.974	148	0.006
UCLA	0.115	148	0.000	0.935	148	0.000

* This is a lower limit of the true meaning.

**Table 4 ijerph-22-01713-t004:** Mann–Whitney U test by gender.

Variable	Gender	N	Mean Rank	Sum of Ranks	Mann–Whitney U	Wilcoxon W	Z	Asymp. Sig. (2-Tailed)
PWI-A	Female	105	80.36	8437.50	1852.500	2887.500	−2.093	0.036
	Male	45	64.17	2887.50				
UCLA	Female	104	71.93	7480.50	2020.500	7480.500	−1.518	0.129
	Male	46	83.58	3844.50				

**Table 5 ijerph-22-01713-t005:** Mann–Whitney U test by living arrangements.

Variable	Living Arrangments	N	Mean Rank	Sum of Ranks	Mann–Whitney U	Wilcoxon W	Z	Asymp. Sig. (2-tailed)
PWI-A	Living alone	50	73.33	3666.50	2391.500	3666.500	−0.433	0.665
	Living with household members	100	76.58	7658.50				
UCLA	Living alone	50	82.47	4123.50	2151.500	7201.500	−1.393	0.164
	Living with household members	100	72.02	7201.50				

## Data Availability

The study dataset is available from the authors upon reasonable request.

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
