# Peer review of "Quality of Life and Loneliness Among Older Adults in Primorsko-Goranska County"

_ijerph, 2025, doi:10.3390/ijerph22111713_

Round 1

Reviewer 1 Report

Comments and Suggestions for Authors

Dear author, I am reviewing your article entitled: " Quality of life and loneliness among older adults in Primorsko Goranska County". Considering that the issue of quality of life and loneliness are important issues in old age, this article has strengths and weaknesses, and I hope that my comments will be effective in improving your work.

  1. The definitions provided in the introduction section regarding aging, especially the first and second paragraphs, are not coherent. In other words, thematic coherence between the paragraphs of the introduction should be observed, which was not the case.
  2. Considering that it is stated in the introduction that various studies, especially meta-analyses and systematic reviews, have been conducted and the relationship between quality of life and feelings of loneliness has been examined and confirmed, what is the necessity of conducting the present study on this similar topic?
  3. What is new and innovative about this work?
  4. The minimum age stated in the Methods and Results section is 63 years old, while the abstract states 65 years and above, which should be corrected.
  5. The sampling method and data collection method are not stated.
  6. The exclusion criteria state that elderly people with Alzheimer's and cognitive impairments were excluded from the study. How did you examine this disease?
  7. The limitations of the study and the applications found should be written at the end of the discussion, whereas in this study, only a portion of it is written at the end of the discussion and most of it is written at the end of the conclusion, which conveys this dispersion of the material.
  8. In the discussion section, cultural differences are mentioned several times to justify the discrepancy between the findings and other studies. It should be made clearer what exactly this difference means.
  9. In the results section, the discussion talks about well-being, not about quality of life. If you were looking at quality of life, are these two concepts the same?
  10. The text has spelling and editing errors and it is recommended that it be rewritten by an English speaker.

Reviewer 2 Report

Comments and Suggestions for Authors

Thank you for the opportunity to review this paper.

STRENGTHS

  1. The introduction and background of the paper are written well and provide a comprehensive understanding of the issues faced by older adults in terms of QoL and loneliness.
  2. The methodology is well explained.

SUGGESTIONS

  1. If, as the author(s) say, there are correlational and longitudinal studies that link QoL with loneliness, please explain in greater detail the academic value that this paper provides to the larger academic community. What is it about the study site that could possibly offer new insight into the experiences of the older adult population in terms of their QoL and loneliness? Are there significant differences between this study cohort and the study cohorts of the aforementioned correlational and longitudinal studies that would make a huge difference in the area of aging studies? Suggestion: Kindly reaffirm the academic value of this article by focusing on the unique value it offers to the wider academic community. 
  2. The findings confirm past research with only nuanced differences. This raises the question: what's new? Could the author(s) provide the larger academic community with indications or suggestions of the findings that further advance our understanding of the the implications of loneliness on QoL among the aging adult population?
  3. The discussion is somewhat general. I would suggest to the author(s) to have a more in-depth take on the results. The discussion section is meant to elevate scholarly understanding of the subject matter, the discussion as it is needs to be elevated in detail, argumentation and raise questions regarding QoL and loneliness among the elderly population. Only confirming, sometimes obliquely, the work of past authors is not enough for a discussion to be considered strong. 

Reviewer 3 Report

Comments and Suggestions for Authors

The article "Quality of life and loneliness among older adults in Primorsko-Goranska County" presents a relevant and timely scientific contribution, addressing the relationship between loneliness and quality of life among the elderly in a specific region of Croatia. The topic is of growing importance in the field of global public health, and the study has three clear merits: (i) its regional focus; (ii) the confirmation of international trends regarding the negative association between loneliness and well-being; and (iii) the analytical nuance in not finding significant differences based on living arrangement, suggesting that the quality of relationships is more decisive than cohabitation.

The article is well-structured, following the classic format. However, for publication in a high-impact journal such as this, we believe there are critical weaknesses that require correction. Let's review them:

Inconsistency in the statistical methodology: There is a flagrant contradiction between the analyses declared in the abstract and those described in the methodology section.

- Abstract: "Statistical analyses included descriptive statistics, Spearman correlation and Mann-Whitney U test...".

- Methods: "Pearson's correlation coefficient was used to analyses the relationship between loneliness and QoL. An independent samples t-test was used to compare QoL and loneliness...".
To resolve this, we propose that the methods section be rewritten to accurately describe the process: the intention to use parametric tests, the verification of data normality (Table 3), and the subsequent and correct choice of non-parametric tests. This transparency is fundamental for methodological rigor.

Absence of formalized research hypotheses: The article defines objectives but fails to formulate clear hypotheses in the introduction. Surprisingly, a hypothesis is mentioned for the first time, and retrospectively, in the conclusion: "It was originally hypothesised that people living alone would report higher levels of loneliness and lower perceived QoL". Therefore, we suggest that explicit hypotheses (H1, H2, H3) be formulated at the end of the introduction, based on the literature review, to guide the reader and structure the analysis.

Ambiguity in the UCLA scale results: The presentation of the mean score for the loneliness scale is confusing, with two different values presented in sequence: "The mean value of the overall UCLA score was 2.52 ± 1.33" and, shortly after, "...while the mean score of the UCLA scale is 34.00, with a standard deviation of 27.14". The authors must clarify what each value represents, opt for a single metric (preferably the mean per item, as the response scale is 1 to 5), and use it consistently.

Incorrect characterization of the sample: The main weakness of the study lies in the description of the sampling, which compromises its external validity. The sample is repeatedly classified as random, which is factually incorrect: "A random sample of 153 adults...". The description of recruitment through "the Belveder-Kozala Pensioners' Club, the ÄŒavle Pensioners' Club..." unequivocally characterizes a convenience sample. The sentence "although a random sample was used in the study, this limits the generalisability of the results" demonstrates a misunderstanding of the concept, as a random sample is precisely intended to increase generalizability. The authors must correct the terminology throughout the manuscript to "convenience sample" and rewrite the limitations section, discussing the true implications of this type of sampling on the generalization of the results.

Regarding the tables, our understanding is that they are appropriate. Figure 1, although illustrative, is redundant given the correlation coefficient presented. The bibliography is current, but the discussion on resilience and self-esteem would benefit from a broader theoretical framework.

In conclusion, we can say that although the topic is relevant and the study's foundation is solid, the methodological and conceptual inconsistencies identified are too significant for acceptance in its current form. Therefore, we recommend the manuscript's acceptance only if the authors undertake a thorough and careful revision of all the points raised.

Round 2

Reviewer 2 Report

Comments and Suggestions for Authors

Thank you for the revisions. The author(s) have covered all relevant points raised by this reviewer. 

Reviewer 3 Report

Comments and Suggestions for Authors

Dear Authors,

I acknowledge that the revised version of your manuscript shows a clear and consistent improvement over the initial submission. The revisions appropriately and thoroughly address the key issues I had previously identified.

The manuscript now demonstrates greater methodological rigor and conceptual clarity. The inconsistencies between the abstract and the methods section have been corrected, with explicit mention of the use of non-parametric tests and verification of data normality. Explicit hypotheses have been formulated at the end of the introduction, giving the study a stronger analytical structure. The presentation of the UCLA scale results has been unified and clarified, and the sample has been correctly characterized as a “convenience sample,” accompanied by a more precise discussion of its implications.

Furthermore, the theoretical discussion has been expanded to include references to resilience, self-esteem, and subjective age, thereby enhancing the interpretative framework of the findings.

Overall, I consider that these revisions reflect a consistent effort toward scientific improvement and adequately address the comments I provided in the first evaluation.

With best regards,
Eduardo Duque